# Rapid spread and population genetics of *Aedes japonicus japonicus* (Diptera: Culicidae) in southeastern Europe (Croatia, Bosnia and Herzegovina, Serbia)

Nele Janssen[1], Nataša Graovac[2], Goran Vignjević[2], Mirta Sudarić Bogojević[2], Nataša Turić[2], Ana Klobučar[3], Mihaela Kavran[4], Dušan Petrić[4], Aleksandra Ignjatović Ćupina [4], Susanne Fischer[1], Doreen Werner[5], Helge Kampen[1], Enrih Merdić [2]*

1 Friedrich-Loeffler-Institut, Federal Research Institute for Animal Health, Greifswald–Insel Riems, Germany, 2 Department of Biology, Josip Juraj Strossmayer University, Osijek, Croatia, 3 Andrija Stampar Teaching Institute of Public Health, Zagreb, Croatia, 4 Laboratory for Medical and Veterinary Entomology, Faculty of Agriculture, University of Novi Sad, Novi Sad, Serbia, 5 Leibniz-Centre for Agricultural Landscape Research, Muencheberg, Germany

* enrih@biologija.unios.hr

## Abstract

The Asian bush mosquito, *Aedes japonicus japonicus* (Theobald, 1901), a potential vector of several pathogens, has recently established in North America and Central Europe. In 2013, it was found on the Slovenian-Croatian border, and during the following years, it emerged in more and more counties of northwestern Croatia. Surveillance of *Ae. j. japonicus* and other invasive mosquito species was subsequently extended both spatially and temporally in Croatia and neighbouring Bosnia and Herzegovina and Serbia. Mosquito collections were conducted in 2017 and 2018, based on adult trapping through dry ice-baited CDC traps and BG-Lure-baited BG-Sentinel traps, larval sampling through dippers and nets, and ovitrapping. *Aedes j. japonicus* specimens from collected samples were subjected to population genetic analysis by comparing microsatellite signatures and *nad*4 DNA sequences between sampled locations and with data previously obtained from more western European distribution areas. *Aedes j. japonicus* immature stages were found at 19 sites in Croatia, two sites in Bosnia and Herzegovina and one site in Serbia. In Croatia, four new counties were found colonised, two in the east and two in the south of the previously known distribution area. A spread of 250 km could thus be documented within five years. The findings in Bosnia and Herzegovina and Serbia represent the first records of *Ae. j. japonicus* in these countries. Genetic analysis suggests at least two introduction events into the surveyed area. Among the locations analysed, Orahovica can be considered a genetic border. The individuals collected west of this point were found to be similar to samples previously collected in the border regions of Southeast Germany/Austria and Austria/Slovenia, while the specimens from more eastern Croatian localities, together with those from Bosnia and Herzegovina and Serbia, were genetically different and could not be assigned to a probable origin. Thus, introduction from Central Europe, possibly by vehicular traffic, into the study

**Data Availability Statement:** All relevant data are within the manuscript and its Supporting Information files.

**Funding:** EM Research in Croatia was supported by Josip Juraj Strossmayer University of Osijek (project ZUP-2018-55); DP in Serbia by the Ministry of Education, Science and Technological Development of the Republic of Serbia (projects III43007 and TR31084), ECDC/EFSA (VectorNet project) and the Secretariat for Urbanism and Environment Protection of Vojvodina Province

**Competing interests:** The authors have declared that no competing interests exist.

**Abbreviations:** ECDC, European Centre for Disease Prevention and Control; EFSA, European Food Safety Authority; WNV, West Nile virus.

area is likely, but other origins, transportation routes and modes of entry appear to contribute. Further dispersal of *Ae. j. japonicus* to other parts of southeastern Europe is anticipated.

## Introduction

The invasive Asian bush or rock pool mosquito *Aedes* (*Hulecoeteomyia*) *japonicus japonicus* (Theobald, 1901) originates from East Asia and the Far East, where it is widespread and even colonises regions with harsh winters [1]. In its native range, the species has a preference for forested and bushy areas, where it is essentially a rock pool breeder [1]. In invaded areas of North America and Europe, larvae develop in rock pools and tree-holes, too, but more frequently and more easily to find, they use artificial containers such as used tyres, rain-water barrels, catch basins, tin cans, bird-baths, roof gutters, flower vases, buckets, etc. [2]. The developmental stages can withstand a wide range of water temperatures but are absent from warm water constantly exposed to sunlight [3, 4]. These facts explain the geographical and altitudinal distribution of the species in its invaded territories, which are usually characterised by moderate climates. Thus, *Ae. j. japonicus* has become widely distributed in North America and Central Europe where it can be found from lowlands to mountainous areas higher than 1,000 m a.s.l. (meters above sea level; e.g., the US Appalachian Mountains and the German Black Forest) [5, 6]. The species is usually absent from areas with temperatures regularly exceeding 30–35˚C [7], although it has succeeded in establishing on Hawaii and in Florida [8, 9].

So far, *Ae. j. japonicus* has not presented itself as an important vector in the field although it has been found infected with West Nile virus (WNV), Japanese encephalitis virus, Cache Valley virus and La Crosse virus [10–13]. Under experimental conditions, it has shown vector-competent for West Nile, Japanese encephalitis, eastern equine encephalitis, La Crosse, Rift Valley, chikungunya and dengue viruses [14–18]. It can be assumed that its actual vector role might temporarily or permanently increase with further spread, rising population densities and numbers of infection sources available as well as climate warming. After all, in some laboratory studies it transmitted WNV even more efficiently than *Culex pipiens* [19] which is particularly alarming before the background of the unprecedented 2018 WNV outbreak in Europe [20] as this suggests widespread and intense virus circulation in natural cycles and, thus, a multitude of infection sources.

*Aedes j. japonicus* is one of the most expansive mosquito species of the world. It is assumed that it is regularly displaced to overseas territories via used tyres in which the females lay their eggs when stored under the open sky and filled with water [21]. Its first evidence outside its native region was in the early 1990s in New Zealand, where it was intercepted but did not succeed in establishing [22]. In 1998, established populations were reported from the eastern USA, and until 2014, this species spread across 34 states in the USA and four states in Canada [2, 8, 23, 24]. In the year 2000, developmental stages were registered for the first time in Europe, in a storage yard of imported used tyres in northwestern France, but were promptly eliminated [25]. Establishment was documented for a population that had been detected in Belgium in 2002 but was not controlled until 2012 as it did not propagate [26]. In Europe, *Ae. j. japonicus* has to date established in Belgium, Switzerland, Germany, the Netherlands, Austria and Slovenia and, more recently, in Hungary, Liechtenstein, Italy, Croatia and Spain [27–29].

The first detection of *Ae. j.japonicus* in Croatia was made in 2013 when eggs were found during a survey for invasive mosquitoes in Krapinsko-Zagorska county, bordering Slovenia

[28] where the species was widely distributed in 2015 [30]. The survey in Croatia included ovitrapping at possible points of entry and in house yards, occasionally complemented by larval collections from cemetery vases. The investigation was continued for another two years and extended to further Croatian counties. In 2014, *Ae. j. japonicus* was detected in northwestern Croatia and in 2015 approximately 100 km eastward of the area of its first record [28]. Within three years, *Ae. j. japonicus* colonised four regional units of northwestern Croatia: Krapinsko-Zagorska, Zagrebačka and Bjelovarsko-Bilogorska counties as well as the city of Zagreb.

Prompted by these findings, a nationwide monitoring programme for invasive mosquito species was initiated in Croatia in early 2016, covering all counties except one in the North-East (Požeško-slavonska). By using ovitraps checked weekly from May to November, new presence data of *Ae. j. japonicus* were obtained from the northern counties Međimurska, Varaždinska, Koprivničko-Križevačka and Virovitičko-Podravska, as well as from the southern counties Karlovačka and Istarska [31]. The continuation of these activities in 2017 produced further distribution data from the eastern county Brodska Posavska [32]. By contrast, no *Ae. j. japonicus* occurrence had been documented before 2017 from Bosnia and Herzegovina and from Serbia, despite recent sporadic (Bosnia and Herzegovina) or extensive invasive mosquito surveillance activities (Serbia).

The present study was meant to follow up on the further spread of *Ae. j. japonicus* in Croatia and check for its occurrence at selected sites close to the Croatian border in neighbouring Serbia and Bosnia and Herzegovina. By genetic analyses of found specimens we attempted to track relationships, origins and transportation routes of populations.

## Materials and methods

### Study areas

The surveillance was carried out in three different areas of Croatia as well as on Bosnia and Herzegovinian and Serbian territories in the border triangle of these three countries (Fig 1).

**Croatia.** Collections were performed in the Slavonian Mountains, Gorski Kotar and the Central Velebit, in the east and south of the known *Ae. j. japonicus* distribution area (Fig 1). All three regions represent mountainous areas characterised by forest vegetation and high precipitation, but are interspersed by villages. Hence, both natural and urban settings were sampled where tree-holes, rock pools and man-made containers were available as suitable breeding sites for *Ae. j. japonicus* in considerable numbers.

The first area is located in Slavonia, northeastern Croatia, in the mountains surrounding Požega Valley. Three transects, along which mosquitoes were collected, crossed Papuk and Požeška Gora mountains, and had a minimum altitude of 177 m and a maximum altitude of 515 m. Papuk is distinguished by its forest richness, dominated by beech and oak, along with maple and ash. The mean annual temperature in this region is 11˚C and the mean annual precipitation 782 mm.

The second area is located in Gorski Kotar, western Croatia, at the foot of the Samarske Stijene rocks (maximum altitude 1,011 m). Gorski Kotar is a plateau with an average elevation of 700–900 m from which mountain peaks rise to up to 1,500 m. Above 1,200 m, the climate is subarctic with a lot of snow, while the lower areas belong to the warm and wet, moderate climate zone. Of particular relevance to the climate are winds, which sometimes reach fierce intensities at higher altitudes. Short and fresh summers and long and harsh winters with lots of snow are characteristic for the sharp mountain climate. This part of Croatia is known for large amounts of precipitation (2,150 mm mean annual rainfall), which are caused by the proximity to the Adriatic Sea and the influence of the high relief. The mean annual temperature is 5˚C. Beech and fir predominate the tree populations.

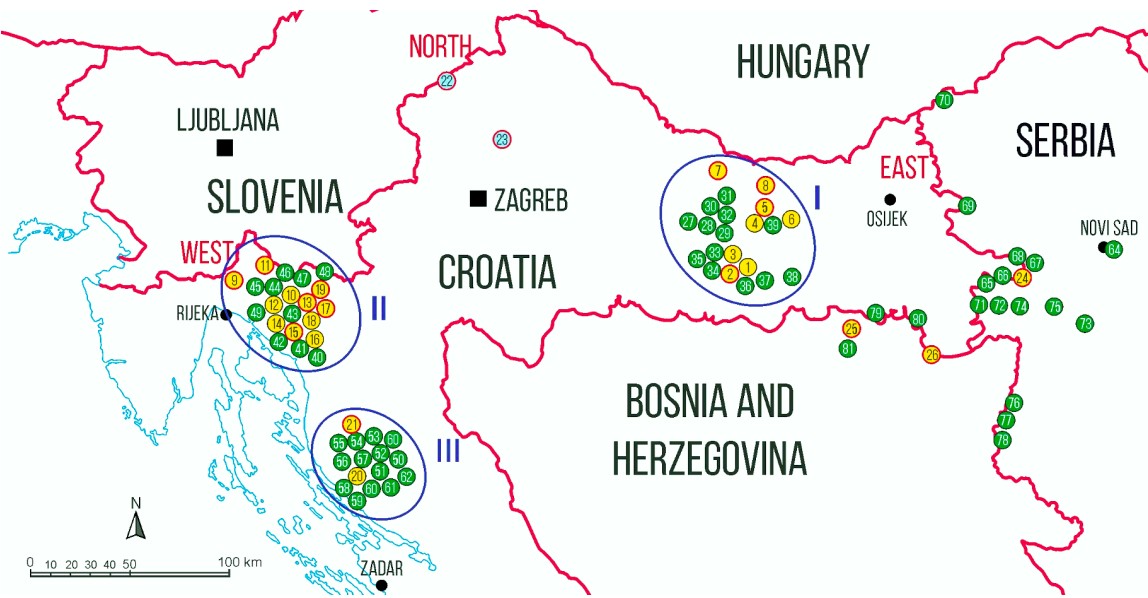

**Fig 1. Study area with monitored locations: Yellow and blue dots mark locations with, green dots locations without *Ae. j. japonicus* findings.** Collection locations considered in the population genetic analysis are marked by dots with red rims. These include two locations, marked by blue dots, found positive for *Ae. j. japonicus* already in the framework of a previous study [28], whereas locations found colonised in the present study and included in the population genetic analyses are marked by yellow dots. Numbers in the dots refer to the compilation of sampled locations in S1 Table, numbers of encirclements to the Croatian study areas (I = Slavonian Mountains, II = Gorski Kotar, III = Central Velebit).

The third area is located in Central Velebit, also in the western part of Croatia and not far from the second study area. Altitudes reach 530 m to 1,000 m. The Velebit mountains build a natural barrier between continental and Mediterranean Croatia. On their peaks, two different climates clash, causing unpredictable weather conditions. The mean annual precipitation in this region is 1,870 mm, while the mean annual temperature is 6.2˚C. The Velebit area is a mosaic of diverse habitats (forests, meadows, rocks, rivers).

**Bosnia and Herzegovina.** Monitoring of invasive mosquito species was carried out in the Posavina region in the northeastern part of the country (average altitude: 90 m a.s.l.), adjacent to Croatian Slavonia. The selected area is located on the banks of two rivers, Sava and Bosna, and close to the Croatian border. The regional climate is generally temperate continental, but often shows extremes. The mean annual temperature in Posavina region is 10˚C and the mean annual precipitation 800 mm. Its vegetation is characterised by deciduous forest, dominated by oak and European hornbeam, but willow and poplar are also quite common. Five municipalities were sampled: Bosanski Šamac, Odžak, Orašje, Brčko and Modriča (Fig 1).

**Serbia.** Invasive mosquitoes were searched for in the northern province of Vojvodina (13 locations) and the western Mačva District (3 locations). Vojvodina is located in the lowest part of the Pannonian Plain (average altitude: 107 m a.s.l.). The mountains surrounding this lowland have a significant impact on its climate, which is continental. Cold winters alternate with hot and humid summers with a huge range of extreme temperatures, featuring inconsistent amounts of rainfall over months, which leads to different levels of aridity. The mean annual temperature is 11˚C and the mean annual precipitation 602 mm.

Mačva is located in the southern edge of the Pannonian basin, between the Cer and Fruška Gora mountains. It has humid subtropical climate with cold winters and warm to hot summers. The altitude ranges from 75 to 95 m and the mean annual precipitation is 774 mm.

The Serbian surveillance activities included eight border crossings to Croatia, three sites close to a border crossing to Bosnia and Herzegovina and one site in Novi Sad, the second-largest city in the country. Two toll and two gas stations located on the motorway E70, which connects Zagreb (Croatia) and Belgrade (Serbia), were also included (Fig 1).

## Mosquito collection

Mosquito collections in Croatia were conducted during July and August 2017 at 58 locations (Fig 1 and S1 Table): 21 in the Slavonian Mountains, 21 in Gorski Kotar and 16 in Central Velebit, once per locality. Collections of developmental stages (larvae, pupae) were carried out utilising plastic dippers or a net, depending on the size of the water bodies which included natural ones, such as ponds, ditches and rock pools, as well as man-made containers, such as tyres, bathtubs, barrels etc. In addition, ovitraps were used at two sites in 2017, and flower vases were checked in a cemetery in 2018. Part of the collected larvae were stored in 100% ethanol for molecular analyses, another part transferred to the laboratory and mounted on slides for a reference collection as described by Merdić et al. [33]. Adult mosquitoes were to be caught using dry ice-baited CDC traps (Bioquip, Rancho Dominguez CA, USA) at 30, and BG-Sentinel traps equipped with BG-Lure (Biogents, Regensburg, Germany) and dry ice as attractants at six of the 58 locations. All traps were operated for at least 12 hours over night, including dusk and dawn. Both larvae and adults were morphologically identified according to Gutsevich et al. [34].

In Bosnia and Herzegovina and Serbia, the surveillance was carried out by ovitrapping (using standard ovitraps of 0.5 l volume and masonite strips as oviposition supports) and larval dipping according to the ECDC (European Centre for Disease Prevention and Control) guidelines for the surveillance of invasive mosquitoes in Europe [35] (Fig 1 and S1 Table). Ovitraps in Bosnia and Herzegovina were positioned from early June to early October 2017 on private properties (n = 15), in cemeteries (n = 4), on border crossings (n = 2), a church garden (n = 1), and on the premises of a petrol station (n = 1), a tyre repair service (n = 1) and a carwash service (n = 1). In Serbia, 95 ovitraps were set up from mid-May to late October (at one location to early December) 2018 in a technical car service in the urban area of Novi Sad (n = 5), on eight border crossings (n = 45), at three petrol stations (n = 13), in a church garden and its surroundings (n = 10), in a forest (n = 5), in a private ethno-village (n = 10) and at two toll stations (n = 7). Distances between the traps were at least 500 m. Fortnightly, the traps were checked and strips with eggs collected. If larvae had already hatched, the water was collected and transferred to the laboratory. On each occasion of checking ovitraps in the five sampled municipalities of Bosnia and Herzegovina, discarded tyres were inspected, too (7 sites). Also, other potential natural and artificial breeding sites close to the ovitraps (up to 20 m) were regularly checked. Both eggs and larvae were reared to adult stages which were then identified according to the key by Gutsevich et al. [34].

Altitude was measured for all locations, as this was meant to be correlated to occurrence of *Ae. j. japonicus*.

The selected study locations (S1 Table) had never been checked for mosquitoes in general or the presence of invasive mosquito species in particular before the present survey. However, mosquito collections carried out sporadically or regularly elsewhere in the study regions (Croatia: since 2016; Serbia: since 2009; Bosnia and Herzegovina: since 2015) did not produce *Ae. j. japonicus* specimens, suggesting its previous absence.

## Population genetic analysis

*Aedes j. japonicus* samples from 16 locations in Croatia, Bosnia and Herzegovina and Serbia (Fig 1) were subjected to population genetic analysis. These included 14 new sites

containing *Ae. j. japonicus* as described here and specimens collected during the present study from two sites previously known to be colonised by the species [28]: Macelj and Konjščina (blue dots in Fig 1).

Following Fonseca et al. [36, 37], the genetic analysis was based on two different approaches: the DNA sequence of the mitochondrial NADH dehydrogenase gene subunit 4 (*nad*4) gene was examined for nucleotide polymorphisms, and microsatellite loci (nuclear DNA) demonstrated to be informative for *Ae. j. japonicus* [38] were analysed for differences in length/number of repetitive motifs.

However, due to poor DNA quality, heteroplasmy (simultaneous presence of two or more haplotypes in the same individual) or microsatellite analysis of certain loci failing in some specimens, not all individuals available could be considered. For that reason and because of a limited number of water containers sampled and of specimens collected per locality, several localities produced only few data.

DNA was extracted using the QIAamp DNA Mini Kit (Qiagen, Germany), following the manufacturer's instructions. Two approaches were applied in parallel: on the one hand, a segment of the mitochondrial *nad*4 gene was sequenced for each specimen, as described by Fonseca et al. [37], Zielke et al. [39] and Janssen et al. [40]. On the other hand, the fragment lengths of seven microsatellite loci (OJ5, OJ10, OJ70, OJ85, OJ100, OJ187, OJ338) were determined, following the protocol of Widdel et al. [38], as modified by Fonseca et al. [36] and Egizi and Fonseca [41]. The results were edited with Geneious 10.2.3 (Biomatters), analysed in STRUCTURE with a Bayesian algorithm [42] (length of burn-in period: 50,000; number of Markov-Chain-Monte-Carlo repetitions after burn-in: 100,000) and evaluated with the software STRUCTURE HARVESTER [43]. Furthermore, a principal coordinate analysis (PCoA) was performed on the microsatellite data using Nei's genetic distance and pairwise $F_{ST}$ values (only localities with more than one specimen) [44].

To get indications on origins and introduction routes, the genetic data were compared to data previously produced for the closest western European *Ae. j. japonicus* populations known, the Southeast German/Austrian and Austrian/Slovenian populations [39, 45].

## Results

### Mosquito collection and identification

*Aedes j. japonicus* was only collected as larvae or eggs. These were obtained from 21 Croatian locations (Fig 1 and S1 Table), where barrels (6 out of 31 checked), tyres (5/20), bathtubs (3/5), ponds (2/9), rock pools (2/2), ovitraps (2/6) and flower vases (1/9) were colonised. The detection sites were located at altitudes from 184 m in Slatinski Drenovac (Slavonian Mountains) to 921 m in Baške Oštarije (Central Velebit). Approximately 52% (11/21) of them were around or above 700 m (Fig 2).

In Bosnia and Herzegovina, *Ae. j. japonicus* eggs were found at two locations close to the Croatian border, Odžak (7 July 2017) and Brčko (9 August 2017), once each (Fig 1 and S1 Table). In Serbia, one single ovitrap positioned on the Serbian-Croatian border in Ljuba was positive for *Ae. j. japonicus* eggs twice (21 August and 15 September 2018) (Fig 1 and S1 Table).

The findings reveal a continuing expansion of the distribution area of *Ae. j. japonicus* in southeastern Europe (Fig 3), with 10 new collection sites in Primorsko-Goranska county, two new sites in Ličko-Senjska county, four new sites in Požeško-Slavonska county and one new site in Osijek Baranja county in Croatia, two new collection sites in Bosnia and Herzegovina and one new collection site in Serbia.

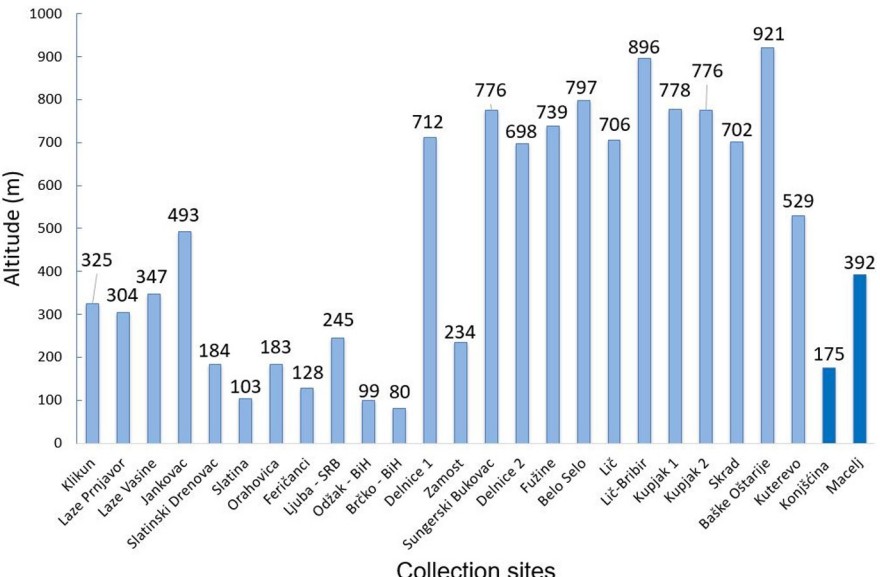

**Fig 2. Altitude of *Ae. j. japonicus* collection locations (SRB = Serbia, BiH = Bosnia and Herzegovina; locations without country code are in Croatia).** Locations with bars coloured in light blue were found positive for *Ae. j. japonicus* in the present study whereas those coloured in dark blue had been found colonised already in the framework of a previous study [28].

## Population genetics

A total of 142 *Ae. j. japonicus* specimens from 16 locations in Croatia, Bosnia and Herzegovina and Serbia were analysed genetically (Table 1).

Analysable microsatellite signatures were produced for 130 individuals, which were grouped into two different clusters according to a Bayesian cluster analysis using the programmes STRUCTURE and STRUCTURE HARVESTER (k = 2, Δk = 24.950; Fig 4). Most of the individuals from Macelj and Orahovica had a relatively high probability of belonging to one cluster, named genotype 1. By contrast, specimens from Zamost, Slatintinski Drenovac, Skrad, Laze Prnjavor, Ljuba and Brčko had a high probability for the second cluster, genotype 2. The other localities were more admixed with intermediate probabilities to belong to one of the two genotypes.

In a second Bayesian analysis (Fig 5), the previously investigated populations from Southeast Germany/Austria and Austria/Slovenia [39, 45] and the populations from southeastern Europe as examined in this study, were jointly analysed. This time, the Bayesian algorithm yielded the highest probability for 13 different clusters (k = 13, Δk = 38,149; Fig 5). The populations from SE-G/AU and AU/SLO as well as those from Delnice 1, Lič, Belo Selo, Skrad, Zamost, Kupjak 1 and Kuterevo turned out to have no preponderate probability of individuals to belong to one or another genotype. Due to this similarity and their geographic origin, they are summarised under the microsatellite signature group 'West Croatia' for the purpose of assessing their genetic makeup with regard to relatedness to specimens from other locations in a PCoA and relating them to haplotype configurations. By contrast, specimens from other locations show a high probability for only one genotype: Konjščina (dark blue), Macelj (light blue), Laze Prnjavor (light green), Orahovica (dark green) and Ljuba (grey). The single specimen from Brčko shows a similar microsatellite signature as the specimens from Odzak, so both are summarised under 'Bosnia and Herzegovina'. The microsatellite signature of the mosquitoes from Slatinski Drenovac and Slatina looks similar to that of 'West Croatia', but is

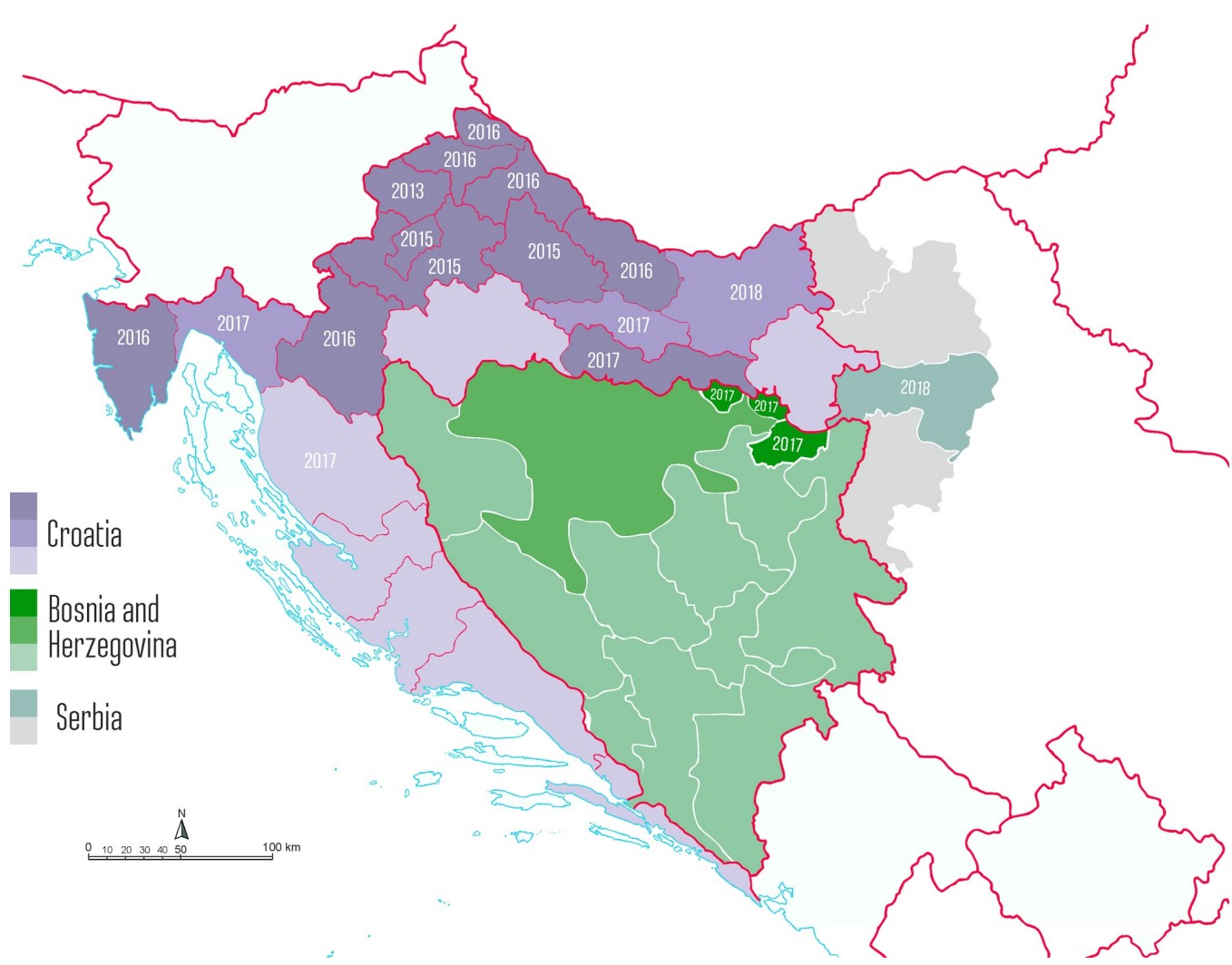

**Fig 3. Results of *Ae j. japonicus* monitoring in Croatia, Bosnia and Herzegovina and Serbia, related to province.** Numbers of years refer to the first detection of *Ae. j. japonicus* (Croatia: dark red–presence of *Ae. japonicus* according to previous studies, light red–presence of *Ae. japonicus* according to this study, orange–monitored, *Ae. j. japonicus* not found; Bosnia and Herzegovina: dark green–presence of *Ae. j. japonicus* according to this study, green–monitored, *Ae. j. japonicus* not found, light green–not monitored; Serbia: dark grey–presence of *Ae. j. japonicus* according to this study, grey–monitored, *Ae. j. japonicus* not found, light grey–not monitored).

clearly influenced by the light green genotype predominating at Laze Prnjavor. These two locations are summarised under the microsatellite group 'East Croatia'. The genetic make-up of the remaining collection sites (Macelj, Konjščina, Orahovica, Laze Prnjavor, Ljuba) showed no similarity and was analysed individually.

The results of the PCoA (Fig 6) confirm the close relatedness of the *Ae. j. japonicus* populations from the Austrian/Slovenian and Southeast German/Austrian border regions with the 'West Croatia' group. The individuals from Konjščina and Ljuba do not seem to be closely related to the other populations.

For 127 samples, *nad*4 sequences could be determined, showing variable base pairs in eight different positions. According to Fonseca et al. [36, 37, pers. comm.] and Zielke et al. [39], these single nucleotide polymorphisms could be assigned to nine *nad*4 haplotypes: H1, H3, H4, H9, H10, H12, H19, H33 and H35 (Fig 7).

**Table 1.** *Aedes j. japonicus*-positive locations from which specimens were subjected to population genetic analysis.

| Country | County | Location | N_Total | N_S | N_M |
|---|---|---|---|---|---|
| Croatia | Primorje-Gorski Kotar | Zamost | 1 | 1 | 1 |
| | | Delnice 1 | 1 | 1 | 1 |
| | | Skrad | 13 | 13 | 13 |
| | | Kupjak 1 | 4 | 4 | 4 |
| | | Belo Selo | 1 | 1 | 1 |
| | | Lič | 11 | 11 | 11 |
| | Lika-Senj | Kuterevo | 5 | 1 | 5 |
| | Krapina-Zagorje | Macelj | 19 | 15 | 19 |
| | | Konjščina | 12 | 10 | 12 |
| | Virovitica-Podravina | Slatina | 4 | 4 | 4 |
| | | Slatinski Drenovac | 4 | 4 | 4 |
| | | Orahovica | 19 | 18 | 18 |
| | Požega-Slavonia | Laze Prnjavor | 20 | 18 | 20 |
| Bosnia and Herzegovina | Posavina | Odzak | 20 | 19 | 9 |
| | | Brčko | 7 | 6 | 7 |
| Serbia | Vojvodina | Ljuba | 7 | 1 | 1 |
| Total | | | 142 | 127 | 130 |

N_Total is the number of specimens processed, independent of successful analysis, N_S the number of individuals examined for *nad*4 sequences and N_M of individuals investigated by microsatellite analysis.

Most common and widespread was haplotype H1 (n = 62), followed by H12 (n = 32). In most of the locations (n = 11), *nad*4 haplotype H1 was dominant, while H12 was dominant in the 'Bosnia and Herzegovina'-group and in Ljuba. In Orahovica, H9 was the dominant *nad*4 haplotype. Orahovica was the only locality with haplotypes H4 and H33 and Macelj the only

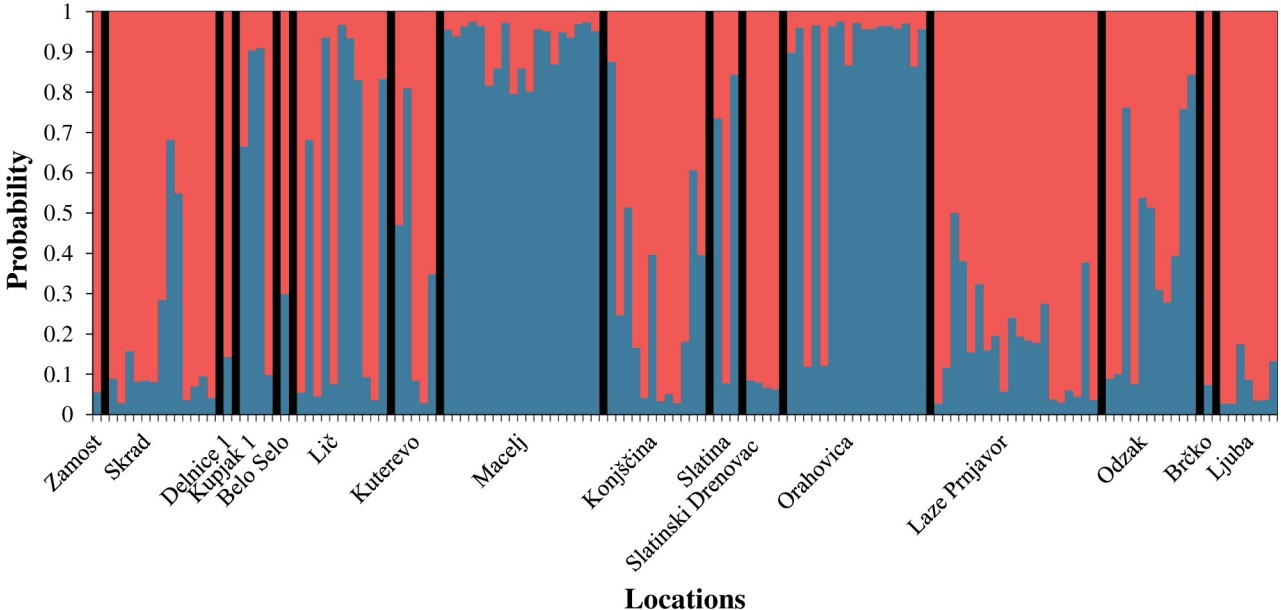

**Fig 4. Bayesian cluster analysis of multilocus microsatellite genotyping (Δk = 2; optimal number of genetic clusters).** Bars represent single individuals, colours the average probabilities of those to belong to one of the two most likely genetic clusters (blue = genotype 1; red = genotype 2).

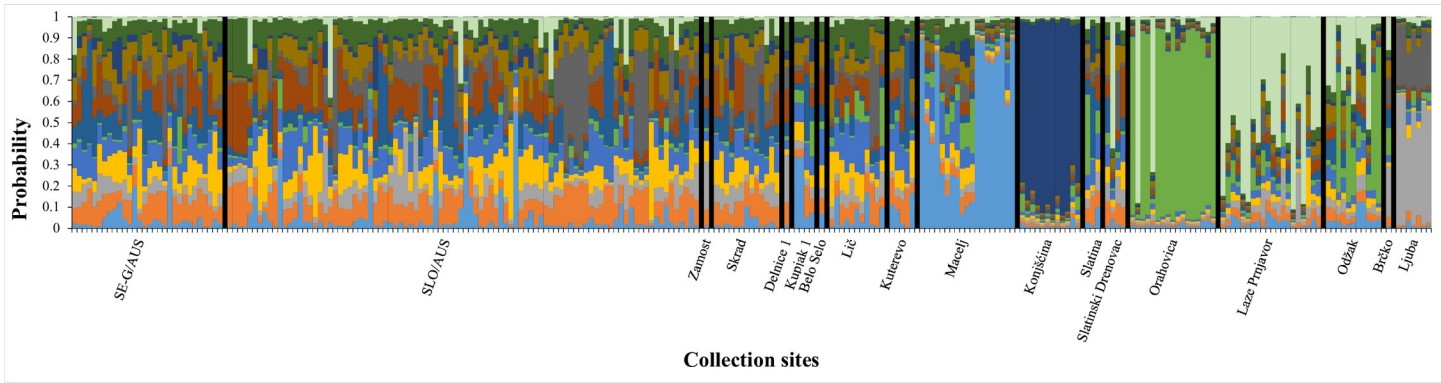

**Fig 5. Bayesian cluster analysis of multilocus microsatellite genotyping (Δk = 13; optimal number of genetic clusters).** Bars represent single individuals, colours the average probabilities of those to belong to one of the 13 most likely genetic clusters as calculated from previously investigated populations from Southeast Germany/ Austria (SE-G/AU) and Austria/Slovenia (AUS/SLO) [42, 43] and of the populations from southeastern Europe as examined in this study.

locality with haplotype H19. At five places, the haplotypes of one single individual could be identified only (Zamost, Delnice 1 and Belo Selo: H1, Kuterevo: H10, Brčko: H12). Fig 8 shows the relative relatedness of the different *nad*4 haplotypes found.

Fig 9 displays the geographic distribution of the most common haplotypes (H1, H5, H9, H10, H12), as related to the microsatellite make-up of *Ae. j. japonicus* at the various

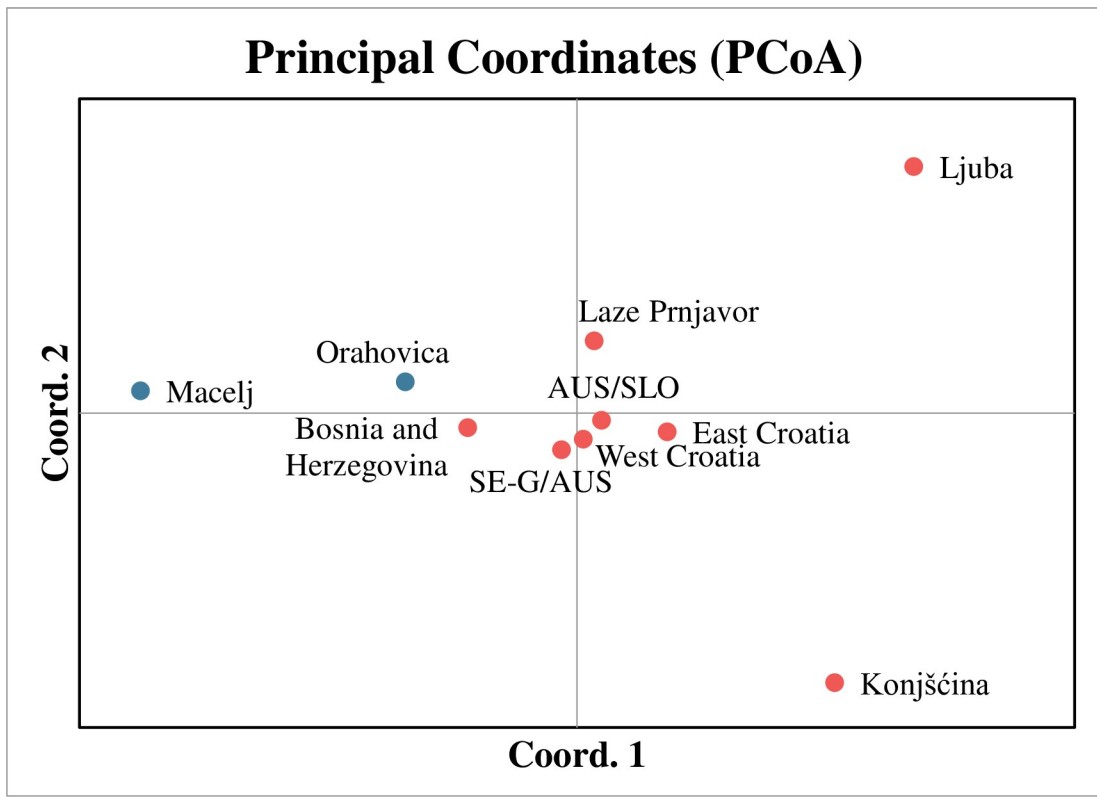

**Fig 6. Principal coordinates analysis (PCoA) plot of pairwise population $F_{ST}$ values for the locations sampled in southeastern Europe in this study and for previously investigated populations from Southeast Germany/Austria (SE-G/AU) and Austria/Slovenia (AU/SLO) [39, 45].**

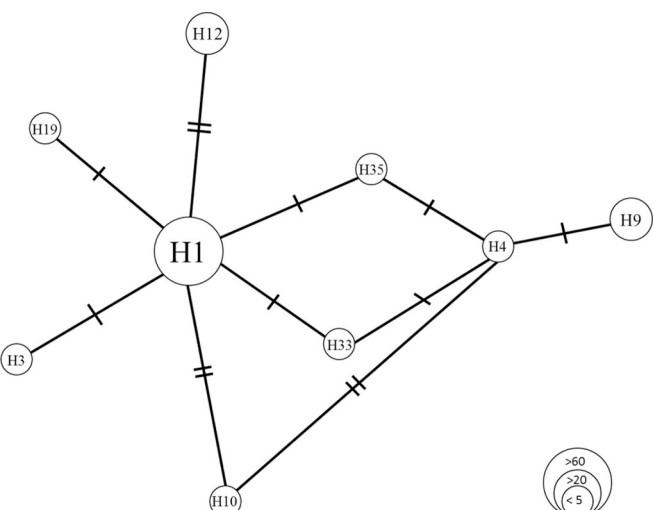

**Fig 7. Frequencies of *Ae. j. japonicus* haplotypes found, based on presorting of populations according to microsatellite clusters.**

**Fig 8. Relative relatedness of determined *nad*4 haplotypes.** The circle sizes represent the number of detected haplotypes, every crossed line represents one transition.

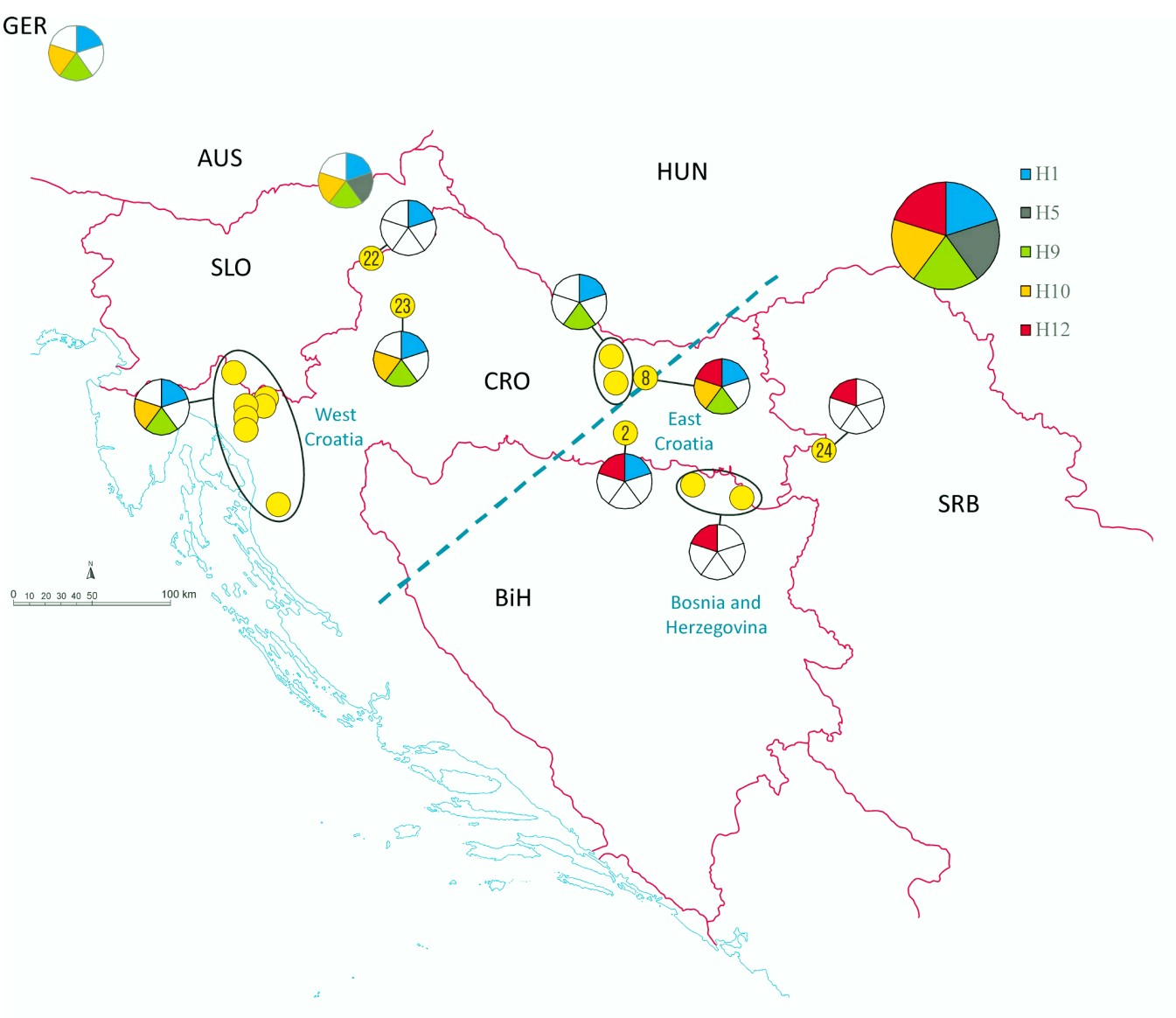

**Fig 9. Geographic distribution of the most common *nad*4 haplotypes H1, H5, H9, H10 and H12 as related to the microsatellite make-up of *Ae. j. japonicus* at the various locations.** In case of absent *nad*4 haplotypes, the pie chart field is white. Numbering of collection sites: 2 = Laze Prnjavor, 8 = Orahovica, 22 = Macelj, 23 = Konjšćina, 24 = Ljuba. Encircled collection locations mark the microsatellite groups 'West Croatia', 'East Croatia' and 'Bosnia and Herzegovina', the dashed blue line may represent a "genetic border".

locations. In the western regions, haplotypes H1, H9 and H10 are very common. Towards the east, haplotypes H9 and H10 become rarer and are finally absent. In the very east, only H12 occurs. The distribution of the *nad*4-haplotyes shows a division of the collection sites into two areas, with Orahovica, displaying the highest diversity of haplotypes, in between. All collection sites east of Orahovica are characterised by a high number of *nad*4-haplotype H12. By contrast, the locations to the west are dominated by H1 and H9. The *nad*4 haplotypes detected in 'West Croatia', 'East Croatia' and 'Bosnia and Herzegovina' do not disagree to the microsatellite signature grouping, but a direct correlation of *nad*4-haplotype and genotype of the microsatellite analysis is not possible.

## Discussion

According to the ECDC, ovitrapping is the method of choice for checking points of entry or small-size areas for presence and absence of invasive mosquito species [35]. To follow up on the spread of *Ae. j. japonicus* on a wider scale, Koban et al. [27] suggest larval sampling, e.g. in cemeteries, allotment gardens or deciduous forest with plenty of tree-holes. Thus, ovitrapping and inspection of cemeteries in urban areas has constituted the main method of monitoring in Croatia since 2013, after *Ae. j. japonicus* had first been recorded in western Croatia [28]. The same is true for Serbia regarding surveillance of *Ae. albopictus* (unpublished data), while invasive mosquito monitoring in Bosnia and Herzegovina was carried out only in 2015 and 2016 in the framework of the ECDC/EFSA VectorNet project. While focusing on higher altitudes, various approaches (adult trapping, larval sampling targeting both natural and artificial habitats, ovitrapping) were used in this study to find out about the further spread of *Ae. j. japonicus* in Croatia and adjacent regions.

The study confirms that trapping of adult *Ae. j. japonicus* is not a sensitive detection method. Experience shows that adults of this species are caught by commonly used mosquito traps only in areas with high population densities, and even in such areas only occasionally [46]. Therefore, adult trapping does not seem to be an appropriate approach when trying to track spreading, since newly occupied areas may be characterised by low population densities for a long time [7]. Larval sampling and ovitrapping are usually more successful [27, 47]. In our study, *Ae. j. japonicus* could be detected by larval sampling in the Croatian collection areas both in natural habitats such as ponds and rock pools, and artificial containers such as barrels, tyres and bathtubs, and by ovitrapping in all collection areas.

Due to previous sporadic or systematic sampling in the study area with no *Ae. j. japonicus* specimens being found, it must be assumed that the findings indicate a relatively new presence of the species. However, detection of invasive species certainly depends on sampling effort which is hard to measure and compare when different methodologies are applied in different regions or at different times. One must be aware that not finding a species does not necessarily mean that the species is absent, particularly at the beginning of its colonisation when population densities are low.

Obviously, population densities in the studied area were high enough for *Ae. j. japonicus* to be detected in numerous places. A high proportion of sampled locations (21 out of 58) turned out positive for *Ae. j. japonicus* larvae in Croatia. Particularly many new findings (11 out of 20) were made in Gorski Kotar area, south of the place where *Ae. j. japonicus* had been recorded first in Croatia. Although the new findings represent the first records in this area, the high number of positive sites suggests that the species had probably arrived much earlier and remained undetected. Population densities, however, that could give hints on the duration of establishment, have not been assessed.

*Aedes j. japonicus* is well adapted to temperate climates, and its eggs are capable of enduring cold and snowy winters as occurring in its endemic home range in northern Japan [34]. In Germany, larvae were found in water as cold as 4˚C [47]. This cold tolerance allows the species to occur at higher elevations. In the southern Appalachians, USA, larvae were detected at altitudes of up to 1,500 m a.s.l. where winter temperatures can reach -18˚C [5], while the species is also prevalent at 1,200 m a.s.l. in the German Black Forest [6]. Higher altitudes in southeastern Europe provide comparable conditions and seem to be readily accepted by *Ae. j. japonicus*, as indicated by the high percentage of breeding sites found around or above 700 m a.s.l. This may indicate that mountains are not necessarily barriers to the spread of *Ae. j. japonicus* and enable establishment and survival of this species, which is adapted to moderate climates, in relatively warm (or even subtropical) regions, e.g. of the Mediterranean.

After *Ae. j. japonicus* had been found in Krapinsko-Zagorska county in 2013 and again in 2014, a continuous spread was observed in Croatia: three surrounding counties were found positive in 2015, and another six, mostly in the north, in 2016. According to the present study, four more Croatian counties are now colonised, two counties in Bosnia and Herzegovina and one in Serbia. Only two counties of inland Croatia plus four coastal ones remain without documentation of *Ae. j. japonicus* (Fig 3). One of the inland counties is Sisačko-Moslavačka county in Central Croatia where *Ae. j. japonicus* is most probably present but has not been demonstrated due to insufficient monitoring. The second inland county is Vukovar-Srijem county in eastern Croatia, which has been included in the national monitoring since 2016 but without findings so far. The negative results are interesting considering the findings in Posavina county in Bosnia and Herzegovina in 2017 and Srem county in Serbia in 2018, which border Vukovar-Srijem county. Most likely, population densities in the latter area are rather low, with specimens escaping detection at many places.

In summary, *Ae. j. japonicus* appears to have quickly spread through Croatia from 2013 to 2018, with evidence of presence from all but six counties and even passing the borders to riparian countries in the east. Precisely, the species was found 250 km east and south of the place of its first record within five years, corresponding to an average dispersal of 50 km per year.

In the USA, three states notified collections of *Ae. j. japonicus* in 1998, New Jersey, New York and Connecticut [48, 49]. It is not clear when the introduction had taken place but, supposedly, several years before. In 2005, *Ae. j. japonicus* emerged in Missouri, 1,800 km away from New York [50]. By 2011, the species was noted in 33 US states, including Hawaii [8]. Thus, the speed of spreading was extremely high in the US, approximately 200 km per year, probably due to the tyre trade business [7, 51]. As this business is not well developed in Croatia, spread might essentially be active there [c.f. 47], supported by passive displacement by trade with, and vehicular transport of, horticulture equipment [c.f. 27].

To obtain clues about migration routes and pathways, individuals of *Ae. j. japonicus* collected from 16 locations in Croatia, Bosnia and Herzegovina and Serbia were genetically analysed for *nad*4 region DNA sequences and microsatellite loci signatures. The most common and widespread *nad*4 haplotype H1 is considered the haplotype from which most of the other existent haplotypes have evolved [36]. Because of its high abundance in the study area, H1 had probably been introduced first. The second most frequent *nad*4 haplotype, H12, differs by two bases from other haplotypes and only occurred in the eastern sampling area (Ljuba, Laze Prnjavor, Orahovica and Bosnia and Herzegovina). In 2013, Zielke et al. [52] had detected this haplotype in the Netherlands. Presumably, it must be attributed to a second introduction into the eastern study area. Haplotype H9 (third most common) was identified 23 times. This haplotype had been found by Zielke at al. [39, 45] in Belgium, the Netherlands and Slovenia. Due to its geographic distribution, which is similar to that of H1, it is likely that these haplotypes were imported to the study area simultaneously. In addition to H12, the haplotypes H3, H4 and H33 were also exclusively found in the eastern locations (Ljuba, Brčko, Odzak, Laze Prnjavor, Orahovica). These must therefore be attributed to introductions from unknown or non-investigated populations rather than from the known populations in Southeast Germany/Austria and Austria/Slovenia.

The collection locality Orahovica shows the highest *nad*4 diversity (six haplotypes). All locations east of this point are dominated by haplotype H12 (in addition to H1), and almost all locations (with more than one individual analysed) west of this point show a high number of H9. Therefore, Orahovica can be considered a *nad*4 haplotype border region, influenced by both eastern and western populations.

Furthermore, the cluster analysis shows a high probability of the eastern locations (Brčko, Odzak, Laze Prnjavor, Ljuba) to belong to genotype 2, just like specimens from Konjščina.

However, the PCoA suggests that these two groups are genetically isolated from each other, which may have been caused by gene flow between these and other, unknown populations. All other haplotypes found in the study area are likely to have evolved from H1, H9 and H12. The genotype signatures of the microsatellites also indicate at least two independent introductions.

Most of the results of the PCoA and the Bayesian analyses correspond well to each other. Accordingly, the microsatellite group 'West Croatia' seems to be more closely related to the populations from SE-G/AUS and AUS/SLO, although comparative analysis of the microsatellite dataset of these populations shows a high number of probable genetic clusters and does not allow substantiated conclusions. Nevertheless, a genetic similarity between these three and the genetic group 'East Croatia' is given, suggesting gene flow may have taken place. Genetically separated from the other populations and of unknown origin are the collection sites Ljuba, Macelj and Konjšćina.

*Aedes j. japonicus* is not considered an important vector in its native distribution area, and evidence for a substantial role in the transmission of disease agents in the field is generally missing. By contrast, experimental data from the laboratory do suggest a vector potential for several viruses of medical and veterinary relevance, including WNV [53]. *Aedes j. japonicus* was found infected with WNV in the field [11], although this does not allow any conclusions on its vector competence since complete homogenised mosquitoes had been examined. In the last decade, mosquito-borne diseases broke out almost every year in Croatia. Human cases of dengue [54], West Nile fever [55] and Usutu fever [56] infections were registered. The highest prevalence is attributed to human neuroinvasive disease caused by WNV, which has emerged in Croatia every year since 2012. While 38 cases of human West Nile fever were noted until 2017, the number of cases accumulated to 51 in 2018 alone [57], and for the first time eight people died (Vilibić-Čavlek, pers. comm). During the same year, Serbia even recorded 415 human cases [57] with 36 deaths [58]. Therefore, the occurrence of another potential vector in this endemic area, such as *Ae. j. japonicus*, may increase the potential risk of WNV transmission.

## Conclusion

The ongoing quick spread through Croatia and the first records of *Ae. j. japonicus* in neighbouring Bosnia and Herzegovina and Serbia underline the strong expansion drive of this invasive mosquito species and its high adaptation to temperate conditions, irrespective of geography. It appears that *Ae. j. japonicus* prefers higher altitudes in Mediterranean countries to find such conditions, but this has to be further elucidated. The close relatedness of the samples collected at localities northwest of Orahovica to remote populations from more western European countries confirm that introduction and spread are mainly mediated by humans, although on a regional scale, active migration does certainly contribute. Given all these supporting factors, it cannot be expected that the spread of *Ae. j. japonicus* will soon come to an end in Europe. In southeastern Europe, a further spread is anticipated at least in temperate climate areas such as mountainous ones.

## Supporting information

**S1 Table. Study locations, collection approaches and outcome.**
(DOCX)

## Acknowledgments

The authors are grateful to Dorothee Scheuch, Greifswald, Germany, for assistance in data analysis, to Natalija Džojić, Dalma Čavčić, Martina Žulj, students of the Department of Biology

at Josip Juraj Strossmayer University of Osijek, Croatia, as well as to Dejan Lučić, Dragan Dondur, Svetozar Bogdanović and Antonije Žunić, Faculty of Agriculture, University of Novi Sad, Serbia, for assistance in the field.

## Author Contributions

**Conceptualization:** Dušan Petrić, Doreen Werner, Helge Kampen, Enrih Merdić.

**Data curation:** Nele Janssen, Mihaela Kavran, Dušan Petrić, Helge Kampen, Enrih Merdić.

**Formal analysis:** Nele Janssen, Mihaela Kavran, Susanne Fischer, Helge Kampen, Enrih Merdić.

**Funding acquisition:** Dušan Petrić, Doreen Werner, Helge Kampen, Enrih Merdić.

**Investigation:** Nele Janssen, Nataša Graovac, Goran Vignjević, Mirta Sudarić Bogojević, Nataša Turić, Ana Klobučar, Mihaela Kavran.

**Methodology:** Nele Janssen, Dušan Petrić, Susanne Fischer, Doreen Werner, Helge Kampen, Enrih Merdić.

**Project administration:** Dušan Petrić, Helge Kampen, Enrih Merdić.

**Resources:** Dušan Petrić, Doreen Werner, Helge Kampen, Enrih Merdić.

**Software:** Enrih Merdić.

**Supervision:** Dušan Petrić, Helge Kampen, Enrih Merdić.

**Validation:** Nele Janssen, Mihaela Kavran, Susanne Fischer, Helge Kampen, Enrih Merdić.

**Visualization:** Nele Janssen, Helge Kampen, Enrih Merdić.

**Writing – original draft:** Nele Janssen, Mihaela Kavran, Doreen Werner, Helge Kampen, Enrih Merdić.

**Writing – review & editing:** Nataša Graovac, Goran Vignjević, Mirta Sudarić Bogojević, Nataša Turić, Ana Klobučar, Dušan Petrić, Aleksandra Ignjatović Ćupina, Susanne Fischer.

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
