## [Decision Letter · Decision Letter 0]

26 Jun 2020

PONE-D-20-14737

Rapid spread and population genetics of Aedes japonicus (Diptera: Culicidae) in southeastern Europe (Croatia, Bosnia and Herzegovina, Serbia)

PLOS ONE

Dear Dr. Merdic,

Thank you for submitting your manuscript to PLOS ONE. After careful consideration, we feel that it has merit but does not fully meet PLOS ONE’s publication criteria as it currently stands. Therefore, we invite you to submit a revised version of the manuscript that addresses the points raised during the review process.

The two reviewers have raised a number of concerns and suggest corrections that should be addressed in a revised version of he manuscript. While no major changes to the work appear to required, clarification of the points raised will signigficantly improve the manuscript.

We look forward to receiving your revised manuscript.

Kind regards,

João Pinto, Ph.D.

Academic Editor

PLOS ONE

Journal Requirements:

Reviewers' comments:

Reviewer's Responses to Questions

**Comments to the Author**

1. Is the manuscript technically sound, and do the data support the conclusions?

Reviewer #1: Yes

Reviewer #2: Partly

2. Has the statistical analysis been performed appropriately and rigorously? 

Reviewer #1: N/A

Reviewer #2: No

3. Have the authors made all data underlying the findings in their manuscript fully available?

Reviewer #1: Yes

Reviewer #2: Yes

4. Is the manuscript presented in an intelligible fashion and written in standard English?

Reviewer #1: Yes

Reviewer #2: No

5. Review Comments to the Author

Reviewer #1: Janssen et al report results of field studies and genetic study of populations of the invasive mosquito Aedes japonicus in 3 Balkan countries. This study provides new data about the spread of that invasive species and its speed, as well as insights about the spread pathways based on genetic similarities. The article is well structured and written, background information is clear and complete, results are discussed in the right way, tables and figures are clear, and references are adequate and up-to-date. Thus we suggest to accept the manuscript for publication with minor revision, by addressing the following comments and suggestions.

A major point is a need of clarification about the aim of the study, which is not clearly expressed to us in the introduction. There you mention in the last paragraph monitoring programmes for invasive mosquito in the countries considered, but later you describe a wide range of mosquito larval habitats (in M&M, Lines 140-141, 158, 160, 178-180, 196) and presence of other mosquito species who do not breed in Aedes invasive mosquito larval habitats (Line 259-262), while these results are not discussed at all. If you aim was to survey Aedes invasive mosquito species only, then, for more clarity and focus, you might omit larval habitats that are not suitable throughout the manuscript (and also omit results from these); same if focusing only Aedes japonicus but in the former case you might also state about findings or not of other invasive species (Ae. albopictus, aegypti, koreicus…).

Abstract

Line 42 and 43: replace ‘by’ by ‘through’

Line 44: replace ‘included in the collections were subjected’ by ‘from collected samples were subject’

Introduction

Aedes japonicus is a taxon that comprises 4 subspecies, but only one of them (Ae. japonicus japonicus) is known to be invasive and to show some vector competence; thus it remain necessary to specify in the introduction about which subspecies you are writing.

Material and methods

Line 139: give low altitude before high (as in the following paragraphs).

Collection areas description in Croatia: you mention artificial habitats only for the third area; are they absent or rare in the others? Please specify briefly.

Considering the short descriptions of potential larval habitat availability; I would suggest to focus here only on those that are suitable for Ae japonicus (providing insight on availability/abundance of tree holes, rock pools and man-made containers) [see major comment above]

Line 202 (and elsewhere in the MS): replace ‘BG Lure’ by ‘BG-Lure’

Lines 204-205 and 221: ID keys of Becker et al. does not include Aedes japonicus! Thus you may have used another key; please specify.

Line 240: suggest to replace ‘Because of this as well as a limited…’ by ‘For that reason and because of a limited…’

Results

Line 259-262: better to cite only species collected in larval habitats suitable for Aedes invasive species, or species caught in traps at same locations [see major comment above]

Line 370: replace ‘characterized’ by ‘characterised’

Discussion

Line 384-385: what do you mean by ‘limited areas’? Small size areas? Please clarify

Line 385: maybe replace ‘For following up? by ‘To follow up’

Line 397: keep ‘even’ together with ‘in such area’, before or after ‘only occasionally’; add ‘adult’ before ‘trapping’

Line 400: ‘Thus’ is not appropriate here; You could replace by ‘In our study’ or ‘Similarly’

Line 479: new paragraph

Conclusion

Line 502 last sentence: why not in all temperate climate areas (by contract to subtropical areas)?

References

All refs: use normal hyphen or en dash between page numbers (harmonise according to journal’s requirements)

Refs 32, 33, 60: please provide English-translated title

Figures and Tables

-

Reviewer #2: Review of “Rapid spread and population genetics of Aedes japonicus (Diptera: Culicidae) in southeastern Europe (Croatia, Bosnia and Herzegovina, Serbia)” by Nele Janssen.

This manuscript presents an analysis of Aedes japonicus in Croatia, Serbia and Bosnia and Herzegovina using microsatellites and a mitochondrial gene (NADH subunit 4). The paper also describes the rapid spread of this invasive mosquito species in southeaster Europe based on collections and correlation with previous reports.

Regarding the spread of Ae. japonicus in the region, some points need more clarification and discussion in the manuscript. It is not clear if surveillance was being taken in those locations before (224:227), and if not, it is hard to conclude about its spread and new collection sites.

Fig 1, Fig 3, Fig 9: scale is missing;

262: information about the other species collected was not provided;

267: Fig 2 could contain values of collected Ae. japonicus;

304 & 317: it is not clear which method was used to determine the best number of K for STRUCTURE analysis;

305: some sections suggest Macelj samples are from this study but Fig2 suggests otherwise;

Fig5: there is no identification for AU/SLO population; maintenance of the order of populations from Fig4 facilitates comparisons between results. I suspect the number of genetic clusters for this analysis is higher than expected and poor conclusions can be done based on this random distribution of genetic clusters found in “West Croatia”, SE-G and AU-SLO populations;

Table 1: review table, Ntotal for Brčko.

6. PLOS authors have the option to publish the peer review history of their article (what does this mean?). If published, this will include your full peer review and any attached files.

Reviewer #1: **Yes: **Francis Schaffner

Reviewer #2: No

---

## [Author Response · Author response to Decision Letter 0]

2 Oct 2020

PONE-D-20-14737 

Janssen et al.

Response to reviewer

(The lines referred to by the authors pertain to the revised manuscript version in the track changes modus.)

Reviewer #1: 

Janssen et al report results of field studies and genetic study of populations of the invasive mosquito Aedes japonicus in 3 Balkan countries. This study provides new data about the spread of that invasive species and its speed, as well as insights about the spread pathways based on genetic similarities. The article is well structured and written, background information is clear and complete, results are discussed in the right way, tables and figures are clear, and references are adequate and up-to-date. Thus we suggest to accept the manuscript for publication with minor revision, by addressing the following comments and suggestions. A major point is a need of clarification about the aim of the study, which is not clearly expressed to us in the introduction. 

Authors: For clarification, we have elaborated on the aim of the study in the last paragraph of the Introduction (lines 117-122).

There you mention in the last paragraph monitoring programmes for invasive mosquito in the countries considered, but later you describe a wide range of mosquito larval habitats (in M&M, Lines 140-141, 158, 160, 178-180, 196) and presence of other mosquito species who do not breed in Aedes invasive mosquito larval habitats (Line 259-262), while these results are not discussed at all.

If you aim was to survey Aedes invasive mosquito species only, then, for more clarity and focus, you might omit larval habitats that are not suitable throughout the manuscript (and also omit results from these); same if focusing only Aedes japonicus but in the former case you might also state about findings or not of other invasive species (Ae. albopictus, aegypti, koreicus…).

Authors: We agree with the reviewer’s comment on mosquito species other than Ae. j. japonicus. As a consequence, we have deleted all parts dealing with those since elaborations and discussions on them would not add to the actual topic of the manuscript (which is Ae. j. japonicus) and would lengthen the manuscript unnecessarily. In this context, we also focussed the description of mosquito breeding habitats on possible Ae. j. japonicus breeding habitats (lines 141-143).

Abstract Line 42 and 43: replace ‘by’ by ‘through’ 

Authors: Has been replaced according to the reviewer’s suggestion (lines 43 and 44).

Line 44: replace ‘included in the collections were subjected’ by ‘from collected samples were subject’ 

Authors: Has been replaced according to the reviewer’s suggestion (lines 44/45).

Introduction 

Aedes japonicus is a taxon that comprises 4 subspecies, but only one of them (Ae. japonicus japonicus) is known to be invasive and to show some vector competence; thus it remain necessary to specify in the introduction about which subspecies you are writing.

Authors: Subspecies (Ae. j. japonicus) has been specified in the title and throughout the manuscript. 

Material and methods 

Line 139: give low altitude before high (as in the following paragraphs). 

Authors: Has been changed according to the reviewer’s suggestion (line 146).

Collection areas description in Croatia: you mention artificial habitats only for the third area; are they absent or rare in the others? Please specify briefly. Considering the short descriptions of potential larval habitat availability; I would suggest to focus here only on those that are suitable for Ae japonicus (providing insight on availability/abundance of tree holes, rock pools and man-made containers) [see major comment above] 

Authors: As mentioned above, we decided to completely ignore all mosquito species other than Ae. j. japonicus in this manuscript. Consequently, we have modified the descriptions of the collection areas, with only mentioning larval habitats suitable for Ae. j. japonicus.

Line 202 (and elsewhere in the MS): replace ‘BG Lure’ by ‘BG-Lure’ 

Authors: Has been replaced according to the reviewer’s suggestion (lines 43 and 210).

Lines 204-205 and 221: ID keys of Becker et al. does not include Aedes japonicus! Thus you may have used another key; please specify. 

Authors: ID key has been replaced by that of Gutsevich et al. (lines 212/213 and 229). 

Line 240: suggest to replace ‘Because of this as well as a limited…’ by ‘For that reason and because of a limited…’ 

Authors: Has been replaced according to the reviewer’s suggestion (line 248).

Results 

Line 259-262: better to cite only species collected in larval habitats suitable for Aedes invasive species, or species caught in traps at same locations [see major comment above] 

Authors: See comments above: For reasons of clarity, we decided to omit mosquitoes other than Ae. j. japonicus completely from the manuscript. 

Line 370: replace ‘characterized’ by ‘characterised’ 

Authors: ‘characterised’ has been corrected everywhere where used in the manuscript. Due to text modifications, it is not present anymore in the line referred to by the reviewer.

Discussion 

Line 384-385: what do you mean by ‘limited areas’? Small size areas? Please clarify 

Authors: Has been changed to ‘small-size areas’ (lines 397/398).

Line 385: maybe replace ‘For following up? by ‘To follow up’ 

Authors: Has been replaced according to the reviewer’s suggestion (line 398).

Line 397: keep ‘even’ together with ‘in such area’, before or after ‘only occasionally’; 

add ‘adult’ before ‘trapping’ 

Authors: The first reads “…, and even in such areas only occasionally” (line 410). Therefore, it seems to be phrased just as suggested by the reviewer. We therefore do not really understand the reviewer’s comment. We had a native speaker check the phrasing, who found it okay.

Second point has been changed according to the reviewer’s suggestion (line 410).

Line 400: ‘Thus’ is not appropriate here; You could replace by ‘In our study’ or ‘Similarly’ 

Authors: ‘Thus’ has been replaced by ‘In our study’ as suggested by the reviewer (line 413).

Line 479: new paragraph 

Authors: A new paragraph was established but was not visible due to the sentence ending at the very end of the line. The respective paragraph starts in line 497 in the revised manuscript version.

Conclusion 

Line 502 last sentence: why not in all temperate climate areas (by contract to subtropical areas)?

Authors: ‘in temperate climate areas’ has been added, and ‘mountainous ones’ has been used as an example for such temperate areas (lines 521/522).

References All refs: use normal hyphen or en dash between page numbers (harmonise according to journal’s requirements) 

Authors: References have been changed according to journal style, using en-dash.

Refs 32, 33, 60: please provide English-translated title

Authors: Titles of references 32, 33 and 59 (formerly 60) have been translated into English language. 

Reviewer #2: 

Review of “Rapid spread and population genetics of Aedes japonicus (Diptera: Culicidae) in southeastern Europe (Croatia, Bosnia and Herzegovina, Serbia)” by Nele Janssen. This manuscript presents an analysis of Aedes japonicus in Croatia, Serbia and Bosnia and Herzegovina using microsatellites and a mitochondrial gene (NADH subunit 4). The paper also describes the rapid spread of this invasive mosquito species in southeaster Europe based on collections and correlation with previous reports. Regarding the spread of Ae. japonicus in the region, some points need more clarification and discussion in the manuscript. 

It is not clear if surveillance was being taken in those locations before (224:227), and if not, it is hard to conclude about its spread and new collection sites.

Authors: For clarification, we have modified the respective passage a little bit (lines 232-236). The very same locations checked for Ae. j. japonicus in the present study had never before been checked for mosquitoes. However, other locations in the same study areas had been ckecked and found negative for Ae. j. japonicus. 

Fig 1, Fig 3, Fig 9: scale is missing; 

Authors: Scale has been added in all three figures.

262: information about the other species collected was not provided; 

Authors: According to reviewer 1, we have decided to delete all information on species other than Ae. j. japonicus from the manuscript.

267: Fig 2 could contain values of collected Ae. japonicus; 

Authors: The addition of collected Ae. j. japonicus specimens would not only produce overload and, thus, confusion to Fig 2 but also provide a delusive picture since the collection approaches (CDC traps, ovitraps, dipping) and the time invested at the various places were not standardised, so numbers of specimens found are not comparable. The information on the numbers of Ae. j. japonicus specimens collected at the various locations is provided in the manuscript nevertheless (Supplementary Table S1).

304 & 317: it is not clear which method was used to determine the best number of K for STRUCTURE analysis; 

Authors: Additional information on this point has been included for clarification (lines 315-318).

305: some sections suggest Macelj samples are from this study but Fig2 suggests otherwise 

Authors: Except for Macelj and Konjščina which had been found colonised by Ae. j. japonicus already in 2013, all locations dealt with in the present study were found positive for Ae. j. japonicus only in the present study. Samples from Macelj and Konjščina used in the genetic analyses of the present study, however, were also collected in the framework of this study (in 2017 as correctly said in Table S1). The incorrect phrasing in the legends to Figs 1 and 2 has been corrected.

Fig 5: there is no identification for AU/SLO population; 

Authors: Fig 5 contained a labelling error of the x-axis which has been corrected.

maintenance of the order of populations from Fig 4 facilitates comparisons between results. 

Authors: Order of populations from Fig. 4 has been transferred to Fig. 5.

I suspect the number of genetic clusters for this analysis is higher than expected and poor conclusions can be done based on this random distribution of genetic clusters found in “West Croatia”, SE-G and AU-SLO populations; 

Authors: The reviewer is right. We have added a sentence to express the limited interpretation power from the random distribution of genetic clusters (lines 491-494).

Table 1: review table, Ntotal for Brčko.

Authors: Ntotal for Brčko has been corrected.

---

## [Editor Report · Decision Letter 1]

12 Oct 2020

Rapid spread and population genetics of Aedes japonicus japonicus (Diptera: Culicidae) in southeastern Europe (Croatia, Bosnia and Herzegovina, Serbia)

PONE-D-20-14737R1

Dear Dr. Merdic,

We’re pleased to inform you that your manuscript has been judged scientifically suitable for publication and will be formally accepted for publication once it meets all outstanding technical requirements.

Kind regards,

João Pinto, Ph.D.

Academic Editor

PLOS ONE
---

## [Editor Report · Acceptance letter]

16 Oct 2020

PONE-D-20-14737R1 

Rapid spread and population genetics of *Aedes japonicus japonicus* (Diptera: Culicidae) in southeastern Europe (Croatia, Bosnia and Herzegovina, Serbia) 

Dear Dr. Merdić:

I'm pleased to inform you that your manuscript has been deemed suitable for publication in PLOS ONE. Congratulations! Your manuscript is now with our production department. 

Kind regards, 

on behalf of

Dr. João Pinto 

Academic Editor

PLOS ONE